# Photocatalytic anti-Markovnikov hydro- and haloazidation of alkenes

Kang-Jie Bian[1,2], Shijin Yu[1,2], Ying Chen [1], Qiming Liu [1], Xiaowei Chen [1], David Nemoto Jr [1], Shih-Chieh Kao[1], Angel A. Martí [1] & Julian G. West [1] ✉

Hydroazidation of alkenes provides a direct entry to alkyl azides, which are prevalent structural motifs in medicine development and chemical biology probes. While direct access to anti-Markovnikov hydroazidation products has seen recent progress, these protocols are restricted to highly oxidative hypervalent iodine reagents or superstoichiometric metal salts under photochemical conditions with moderate olefin generality. Thus, the development of a mild, catalytic, redox-neutral hydroazidation with high anti-Markovnikov regioselectivity compatible with diverse classes of alkene remains challenging. Here we report a photocatalytic anti-Markovnikov hydroazidation of alkenes enabled by cooperative ligand-to-metal charge transfer (LMCT) and hydrogen atom transfer (HAT). Critical to this protocol is the use of ligand to achieve efficient visible-light induced homolysis of iron azide species and the cooperation with thiol catalysts to promote this redox-neutral process and address previous challenging substrates. Additionally, the photocatalytic system enables a regioselective haloazidation via a tandem LMCT/ halogen atom transfer (XAT) process with judicious choice of halogenating reagents. Preliminary mechanistic studies support a radical nature of this cooperative system and suggest it to be a powerful manifold in olefin hydro- and difunctionalization.

Alkyl azides are prevalent structural motifs among pharmaceuticals, varied synthetic intermediates, and chemical biology probes[1–5]. This synthon can also serve as a compelling nitrene precursor and be leveraged for the construction of valuable carbon-nitrogen bonds of pivotal importance in natural product synthesis and medicinal chemistry[6–10]. Driven by these applications, hydroazidation of olefins presents one of the most direct strategies to access these useful building blocks from abundant feedstock hydrocarbons. Early approaches leveraged acid-mediated Markovnikov addition of explosive and gaseous $HN_3$ to alkenes and proceeds through a carbocation pathway (Fig. 1)[11,12]. Toward avoiding this dangerous reagent, modern protocols developed by Carreira[13,14] and Boger[15] employing metal-hydride atom transfer (MHAT) provided a general and mild route to deliver the same Markovnikov substituted/branched alkyl azides products using alternative azide donors. However, access to linear, anti-

Markovnikov alkyl azide product via these acid-mediated or MHAT mechanisms is unfavorable due to the intermediacy of less stable primary carbocations or alkyl radicals. Indeed, the anti-Markovnikov hydroazidation of alkenes usually requires multistep syntheses[9] and/or stoichiometric use of boranes or electrophilic azide sources[16,17] that resulted in low step and/or atom economy, and constrained generality, presenting challenges in direct anti-Markovnikov hydroazidation.

Toward overcoming these challenges in anti-Markovnikov hydroazidation, several distinct mechanistic approaches have been explored. Chiba, Gagosz, and coworkers leveraged the hypervalent iodine reagent azidobenziodoxlone (Zhdankin reagent) to achieve anti-Markovnikov hydroazidation via a radical addition/1,5 HAT/oxidative debenzylation cascade. While this reaction demonstrated the feasibility of anti-Markovnikov hydroazidation via a radical mechanism, this particular approach is only amenable to specific substrates

[1]Department of Chemistry, Rice University, Houston, TX, USA. [2]These authors contributed equally: Kang-Jie Bian, Shijin Yu. ✉e-mail: jgwest@rice.edu

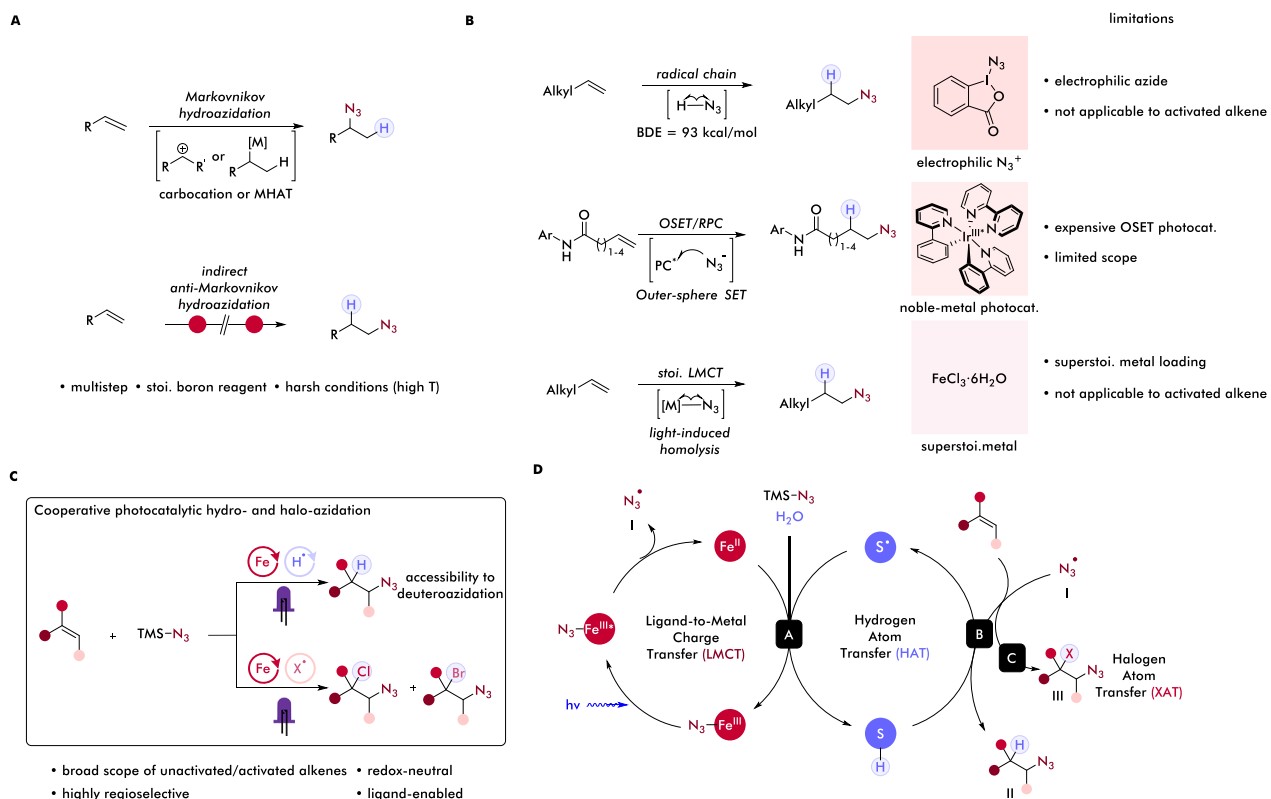

**Fig. 1 | Previous strategies for anti-Markovnikov hydrochlorination and the design of cooperative iron and thiol catalysis for hydrochlorination of unsaturated hydrocarbons. A** Traditional Markovnikov hydroazidation via acid-mediated pathway or metal-hydride catalysis; multistep synthesis to access anti-Markovnikov selectivity. **B** Recent catalytic anti-Markovnikov hydroazidation reactions. **C** This work: Cooperative photocatalytic hydro- and haloazidation of alkenes. **D** Proposed mechanism of photocatalytic anti-Markovnikov hydro- and haloazidation enabled by cooperative ligand-to-metal charge transfer and hydrogen atom transfer or halogen atom transfer. OSET outer-sphere single electron transfer, RPC radical polar crossover.

designed for the full cascade process[18]. Yu and coworkers took a complementary noble-metal based photocatalysis approach to achieve hydroazidation of unsaturated aryl amides via outer-sphere single electron transfer (OSET) of azide, enabling hydroazidation though exhibiting moderate scope and requiring an expensive iridium photocatalyst[19]. By exploiting the reactive nature of hypervalent iodine reagents, which undergoes facile homolytic cleavage of I-N₃ bonds and promote radical chain pathways in protic solvent, Xu[20] and Liu[21] independently reported elegant protocols to achieve direct anti-Markovnikov hydroazidation of alkenes. While synthetically enabling, the adoption of these highly energetic and oxidative reagents could still present barriers to oxidatively-labile functionalities. Additionally, this radical chain pathway is driven by the thermodynamically-favored hydrogen atom transfer (HAT) of in situ generated HN₃ (BDE = 93.3 kcal/mol) to non-stabilized carbon-centered radical intermediates, making chain propagation less favorable with activated alkenes such as styrene due to endogenic HAT process to form a weaker bond (~86 kcal/mol for ethylbenzene)[22]. Therefore, a general anti-Markovnikov hydroazidation that can accommodate a diverse range of activated and unactivated alkenes and potentially reactive functional groups under mild conditions remains underdeveloped. Moreover, such system could also serve as a foundation for complementary functionalizations proceeding via open-shell azidyl radicals, such as haloazidation of alkenes, that would allow facile incorporation of two reactive handles. Previous protocols for haloazidation are largely confined to the in situ preparation of halogen azide in the presence of molecular halogen, affording poor regioselectivity and limited substrate scope[23,24], and/or the requirement of directing group or specific

substrate class for achieving high regioselectivity control[25,26]. We hypothesize that many of these challenges might be overcome by an azidyl radical generation that is mild and amenable to broader functionalization. Taking these examples together, we recognized a mechanistically distinct azidyl radical generation method that avoids the usage of exogeneous oxidant and/or electrophilic/oxidative azidyl radical precursors might permit the efficient anti-Markovnikov hydroazidation and regioselective haloazidation of diverse alkenes, serving as a powerful and unified tool for accessing these useful azidated structural motifs.

Towards azidyl radical generation, previous methods have employed hypervalent iodine reagent BI–N₃, which is able to undergo facile homolytic cleavage of the I–N bond due to its low bond dissociation energy (27.8 kcal/mol)[27,28] or conventional photoredox approaches using noble-metal based photocatalysts, proceeding through a biomolecular, redox-potential matching, OSET oxidation of azide to azidyl radical[29,30]. A complementary approach is inner-sphere single electron transfer (ISET), with electron transfer occurring directly between a coordinated substrate and a metal. Light-driven ligand-to-metal charge transfer (LMCT) is one such ISET process that enables coordinated anionic, X-type ligands to be selectively oxidized to their radical form using metal catalysts of moderate ground state oxidation potential and short excited state lifetimes due to this intramolecular, excited state mechanism[31,32]. Gaining inspiration from Kochi's pioneering work[33] in LMCT of cupric chloride to generate chlorine radical for C-H chlorination and alkene dichlorination and rapid expansion of this LMCT reactivities to other earth-abundant metals in recent years[34–56], our group[57] and Shi and coworkers[58] independently,

developed the first stoichiometric photochemical diazidation of alkenes enabled by iron LMCT and radical ligand transfer as well as our subsequent work making this protocol photocatalytic[59], where we postulated the iron-azide species formed in situ would undergo photolysis to release azidyl radical. Very recently, the azidyl radical generation from iron LMCT reactivity was elegantly adopted by Carreira and coworkers to achieve the non-catalytic iron-mediated photochemical hydroazidation of alkenes[60]. The protocol can accommodate a broad array of diversely substituted alkenes, with preliminary mechanistic studies showing water from iron hydrate as H-atom source. While demonstrating synthetic versatility, the super-stoichiometric use of the iron salt still raises issues in sustainable synthesis, with a potential additional hazard through forming high concentration of metal-azide species in the system. Additionally, similar to previous hydroazidation protocols, activated alkenes such as styrene or α-heteroatom alkenes were not tolerated, possibly due to thermodynamically less favored C–H bond forming HAT (benzylic C-H, BDE ~ 86 kcal/mol; H-N₃, BDE = 93 kcal/mol)[61]. We envisioned that this thermodynamic mismatch might be overcome via HAT to the carbon-centered radical intermediate occurring from a redox-active hydrogen donor that can not only initiate facile HAT facilitated by BDE-driven thermodynamics and polarity-matching kinetics, but also enable reoxidation of lower valent iron formed after LMCT back to the high-valent photoactive state. We have obtained preliminary support for this design through development of iron LMCT/thiol HAT cooperative photocatalysis systems to achieve protodecarboxylation of carboxylic acids[62] and hydrofluoroalkylation of alkenes[63] with both reactions including these key elementary steps. From this, we hypothesized that this cooperative LMCT/HAT system could provide a redox-neutral, photocatalytic strategy to anti-Markovnikov hydroazidation of alkenes.

We proposed the reaction beginning with the photo-induced cleavage of a pre-associated $N_3.Fe^{III}$ species to generate azidyl radical **I** and lower valent iron species. The azidyl radical can then add to an alkene, forming a carbon-centered radical intermediate. Redox-active thiol could subsequently function as hydrogen atom donor to this intermediate via HAT to furnish anti-Markovnikov hydroazidation product **II** with concurrent generation of thiyl radical (step B). Finally, thiyl radical can further oxidize lower valent iron back to its photoactive state, followed by protonation by $H_2O$ co-solvent to form the active thiol catalyst, closing both catalytic cycles (step A). Furthermore, by replacing $H_2O$ with cheap $D_2O$, the deuteroazidation isotopologue is expected to form, providing a unified route to prepare D-labelled azidation products. The intermediacy of carbon-centered radicals can also be utilized in conjunction with somophiles such as NCS or NBS via halogen atom transfer (XAT) to achieve a regioselective haloazidation, providing a versatile protocol to access varied haloazidated products.

Herein, we report a general, photocatalytic anti-Markovnikov hydroazidation of alkenes enabled by cooperative LMCT and HAT. Following our original discovery of azidyl radical generation via LMCT of iron-azide species, we combine this chemistry with HAT of redox-active thiol to permit anti-Markovnikov hydroazidation process to proceed with no additional redox reagents or superstoichiometric mediator. Critical to success of this reaction is presence of a supporting ligand for iron, where the efficiency is improved compared with the "ligand-free" condition and milder, lower energy visible light (427 nm) irradiation can be used instead of high-energy near UV irradiation. Zuo and coworkers reported a pioneering study of iron terpyridine catalysis utilizing LMCT to achieve highly efficient aerobic carbonylation of methane, showcasing the unique photochemical properties of ligated iron complexes in LMCT processes[64]. Notably, this cooperative protocol not only accommodates previously unreactive substrates classes but also allows D-labelled isotopologue preparation, dramatically expanding the applications of this reaction.

Additionally, common halogenation reagents can be used in place of thiol to perform efficient XAT to sequester alkyl radical intermediates and achieve regioselective chloro- and bromo-azidation of alkenes. Together, these cooperative strategies enable facile preparation of diverse alkyl azides under mild and sustainable iron photocatalysis, introducing a powerful new tool for azidofunctionalization of olefins.

## Results

### Reaction design and optimization

We began exploring the possibility of cooperative anti-Markovnikov hydroazidation by using pent-4-en-1-yl benzoate as our model substrate, combined with trimethylsilyl azide (TMSN₃) as the anionic azide source, catalytic loading of earth-abundant iron salt, bi- or tridentate ligand, redox-active thiol, 19:1 HCCl₃/water as solvent, and irradiation with 427 nm LEDs at room temperature. Excitingly, the inexpensive Fe(NO₃)₃·9H₂O (10 mol%) and tridentate terpyridine ligand (terpy, 10 mol%) in cooperation with 4-F-thiophenol (10 mol%) could facilitate the production of highly anti-Markovnikov product **1** in 76% isolated yield, supporting our hypothesized cooperative catalysis scheme (Table 1, entry 1). To differentiate our system from previous protocols that are based on radical chain propagation of HN₃ or stoichiometric use of iron hydrate and provide support that ligand is indispensable in achieving high efficiency, we tested 30 mol% of iron(III) nitrate nonahydrate in the absence of both ligands and thiol co-catalyst under 390 nm and observed drastically decreased yields, supporting the reaction is not likely to operate via the same radical chain mechanism seen in previous reports (Table 1, entry 2). Furthermore, the ligated iron complex exhibited a notable red-shift for effective illumination wavelength, with 390 nm irradiation showing significantly reduced reactivity compared to 427 nm (Table 1, entry 3, for more details, see supporting information). We next observed that unligated iron exhibits almost absent reactivity at 427 nm (Table 1, entry 4), further emphasizing the enabling effect of ligand. Toward further exploring the influence of ligand on the system, we tested a variety of polypyridyl ligands. While bidentate and tridentate pyridine-based ligands showed drastically increased reactivities compared with entries without ligands, the tridentate 2,2′:6′,2″-terpyridine was found to be the optimal ligand of those tested (Table 1, entry 5 and supporting information). Next, we found the identity of thiol to have a strong influence on reaction efficiency, finding electron-rich aryl thiol showed unsatisfying efficiency and electron-deficient aryl thiol or disulfide could provide improved yields, with 4-F-thiophenol performing most efficiently (Table 1, entries 6 and supporting information). Interestingly, non-protic solvents such as acetonitrile and acetone, which are effective for diazidation in our previous work, gave trace hydroazidation product (Table 1, entries 7), leading us to continue using the initial mixed CHCl₃/H₂O solvent system. Higher intensity blue light (50%) and extended reaction time allowed the product yield to be improved to 82% (Table 1, entry 8). Lastly, control experiments indicated no conversion in the absence of iron or light, supporting azidyl radical generation from light-induced homolysis of iron-azide species (Table 1, entries 9–10).

### Scope of anti-Markovnikov hydroazidation of alkenes

With the optimized conditions in hand, we next investigated the functional group compatibility of our photocatalytic hydroazidation (Fig. 2). First, simple, aliphatic alkenes (2, 3) and the tosylate protecting group (6) were tolerated, providing corresponding primary alkyl azides in moderate to high yields. Substituted arenes (4, 5) with electronically distinct functionalities were compatible with this hydroazidation system. 2,2,2-Trichloroethoxycarbonyl (Troc) (7) and N-methylated sulfonamides (8) were also tolerated, giving good yields of hydroazidation products, further demonstrating compatibility of this reaction with synthetically and medicinally relevant functionality. Next, a broad range of heterocycles such as thiophene

**Table 1 | Optimization of catalytic hydrochlorination using cooperative iron and thiol catalysis**

| Entry | Deviation from standard conditions | Yield (%)[a] |
|---|---|---|
| 1 | None | 78(76) |
| 2[b,c] | 30 mol% Fe(NO$_3$)$_3$·9H$_2$O, w/o ligand or HAT cat. | 20 |
| 3[b] | 10 mol% Fe(NO$_3$)$_3$·9H$_2$O, w/ ligand | 22 |
| 4 | 10 mol% Fe(NO$_3$)$_3$·9H$_2$O, w/o ligand | Trace |
| 5 | Pyridine-based bidentate and tridendate ligands | 12–66 |
| 6 | Electron-rich, -deficient thiophenol and phenyldisulfide | Trace-66 |
| 7 | CH$_3$CN, Acetone, EA, DCM, DCE, THF, PhCF$_3$ | Trace-44 |
| 8 | 48 h, 50% 427 nm | 82 |
| 9 | No iron salt and ligand | ND |
| 10 | No light | ND |

Reaction conditions: alkene (0.1 mmol, 1.0 equiv.), TMSN$_3$ (4.0 equiv.), Fe(NO$_3$)$_3$·9H$_2$O (10 mol%), terpyridine (10 mol%), 4-F-thiolphenol (10 mol%), and HCCl$_3$/H$_2$O (9:1, 0.1 M), 36 h, RT, 427 nm Kessil LED. Entries 2–4, 6 were screened with 3.0 equiv. of TMSN$_3$.

[a]$^1$H NMR yield is determined by using CH$_2$Br$_2$ as an internal standard. Isolated yield in the parentheses.

[b] 390 nm Kessil LED.

[c] HCCl$_3$ (0.1 M) as solvent.

(9), pyrrole (10), N-phthalimide (11), tetrahydropyran (12), and benzofuran (13) also underwent efficient hydroazidation under our photocatalytic conditions, overcoming limitation in protocols exploiting highly oxidizing conditions. We next sought to test substrates bearing strained ring structures and acid-, nucleophilic substitution-, and redox-labile functionalities. After observing high yields of terminal azides featuring strained ring such as tetramethylated cyclopropane (14) and oxetane (15), we were excited to see smooth transformation of an alkene substituted with an alkyl bromide handle which remains intact without any azide displacement, providing dual (pseudo)halogenated (Br- and N$_3$-) alkanes in 72% yield (16). Importantly, various redox-labile functionalities including ester (17), sulfide (18), and alcohol (19) all afforded corresponding primary azides in moderate to good yields, showcasing a powerful solution to accessing azidated molecules with these reactive functional groups, which suggests another advantage of our mild photocatalytic system in comparison with protocols using highly oxidizing hypervalent iodine reagents. We next shifted our investigation to different alkene substitutions, finding 1,1-disubstituted alkenes (21-24) behave well, without forming detected Markovnikov products (> 20:1 r.r.). Similarly, 1,2-disubstituted alkenes (25, 26) and disubstituted cyclized alkene (27) both demonstrated encouraging reactivities and provided expected near equal quantities of regioisomers. Trisubstituted alkenes (29, 30) showed excellent compatibility to our system, giving moderate to high yields of corresponding alkyl azides in exclusive anti-Markovnikov regioselectivities. More sterically hindered tetrasubstituted alkene (31) also gave good yield of corresponding product in a mixture of both regioisomers. Finally, R-carvone, a substrate containing two chemically distinct olefinic functionalities (32), was tested to investigate the chemoselectivity of this system and expectedly provided reaction at the non-conjugated site in exclusive anti-Markovnikov regioselectivity.

To further illustrate the importance of this anti-Markovnikov hydroazidation and showcase these alkyl azide handle could be useful for chemical biology probe development, we next endeavored to explore a wide array of alkenes derived from commercially available pharmaceutical ingredients (APIs) and natural products to evaluate viability of our method in late-stage functionalization[65], providing a direct and versatile route to diverse alkyl azides featuring medicinally-relevant motifs. Olefins derived from the NSAID Ibuprofen (33), Oxaprozin (39), lipid-lowering Clofibric acid (34), Benzafibrate (37), sulfonamide-containing Probenecid (35), and natural product Flavone-derived alkene (36) provided hydroazidated products smoothly in moderate to good yields (43%–76%). Notably, Proline-derived alkene (38) featuring low-BDE sp$^3$ α-C-H bonds adjacent to a heteroatom that are the liability for other hydroazidation methods, performs well in our reaction, further demonstrating its chemoselectivity. Alkenes derived from natural product L-Menthol (40), ketal-protected monosaccharide (41), and the complex steroid 18β-Glycyrrhetinic acid (42) behaved well and afforded hydroazidated products in useful yields. Taken together, the high catalytic reactivity, regioselectivity, and broad tolerance of our hydroazidation to structurally diverse alkenes recommend our method as a powerful route to efficiently install azide handles in alkenes featuring biologically active fragments that could help accelerate medicinal chemistry campaigns.

To further differentiate our cooperative LMCT/HAT system from the established radical chain pathways and stoichiometric metal-mediated methods and address challenging substrates[20,21,60], we then investigated problematic olefins in previous protocols. A common incompatible substrate class is styrene-type activated alkenes and we reason this could be correlated to thermodynamically less favored HAT process between H-N$_3$ and benzylic radicals (benzylic C-H, BDE = 86 kcal/mol; H-N$_3$, BDE = 93 kcal/mol) when proceeding via a radical chain pathway. We envisioned that leveraging an aryl thiol HAT cocatalyst (PhS-H, BDE = 84 kcal/mol)[66] should provide more thermodynamic incentive for C-H bond formation and permit anti-Markovnikov hydroazidation of activated alkenes. Excitingly, we found this to be the case, with diversely substituted styrene-type alkenes affording anti-Markovnikov products (44–46) in moderate to good yields. Additionally, this cooperative system can also be applied to activated alkenes adjacent to halogen (47) and heteroatom (48), indicating a general solution for working with these classes of substrates. Furthermore, alkenes substituted with sensitive functionalities such as carboxylic acid (43) or carbazole (49) that are categorized as incompatible examples in previous work were also tolerated under this

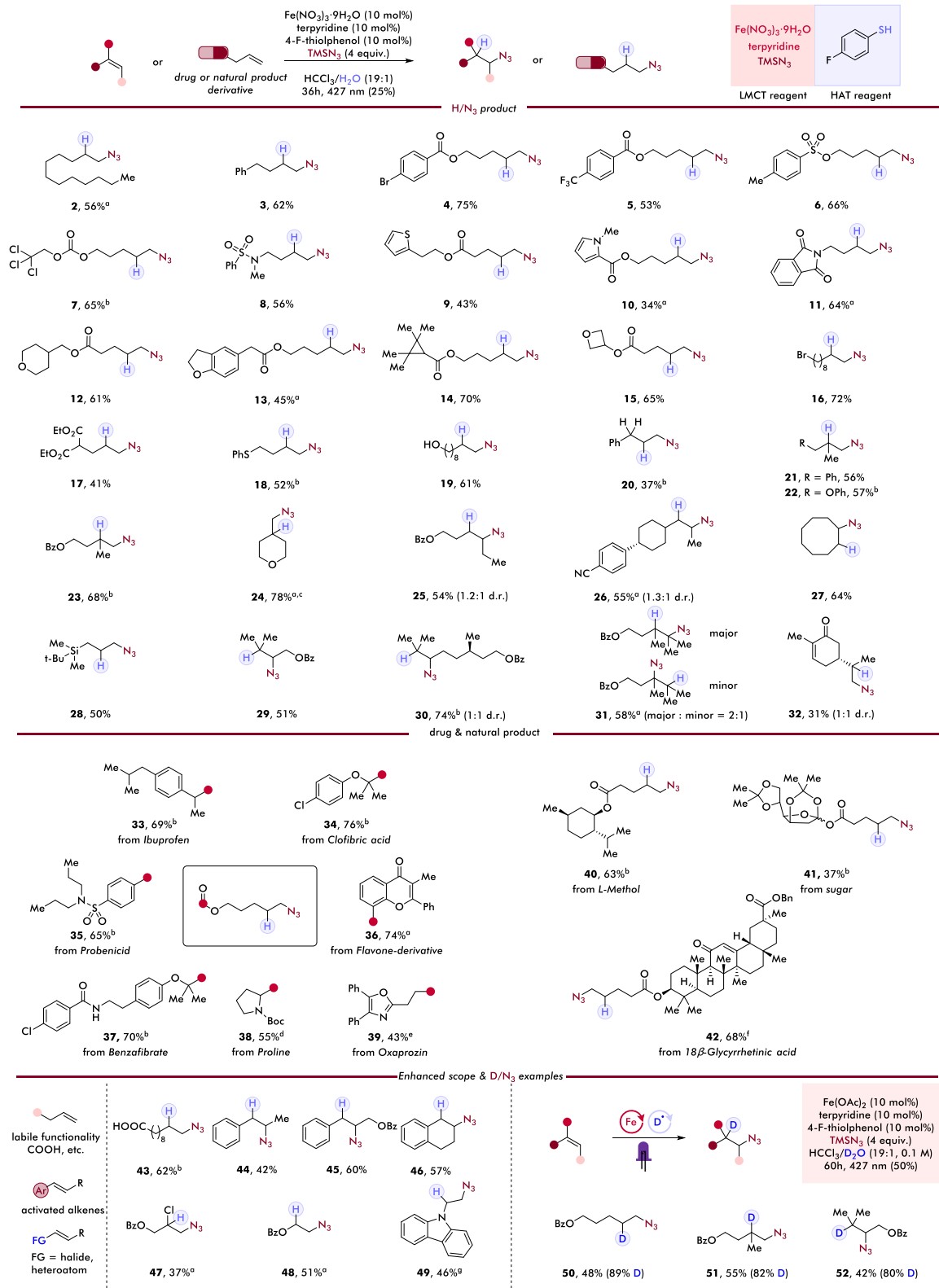

**Fig. 2 | Scope of hydroazidation of alkenes.** Reaction conditions: alkene (0.1 mmol, 1.0 equiv.), TMSN₃ (4.0 equiv.), Fe(NO₃)₃·9H₂O (10 mol%), terpyridine (10 mol%), 4-F-thiophenol (10 mol%), and HCCl₃/H₂O (19:1, 0.1 M), 24 h, RT, 427 nm Kessil LED (25%). [a] With 50% 427 nm light, 60 h. [b] With 25% light, 60 h. [c] NMR yield is determined due to volatile nature. [d] With 50% light, 48 h. [e] With 50% 427 nm light, 72 h. [f] With 25% 427 nm light, 96 h. [g] With Fe(OAc)2, isolated yield indicates anti-Markovnikov product yield.

photocatalytic condition[60], providing corresponding products in moderate to good yields (46–62%). The enhanced tolerance of our protocol, in conjunction with other synthetic advantages in this photocatalytic strategy, demonstrates it to be a general and mild pathway to access anti-Markovnikov hydroazidation.

In our proposed mechanism, thiol transfers a hydrogen atom originating as a proton from $H_2O$ via HAT to the carbon-centered radical, resulting in C–H bond formation. Accordingly, this origin suggests that simply replacing $H_2O$ with $D_2O$ could result in deuterium atom transfer[67,68] to access D-labelled alkyl azides. Implementing this design, we carried out preliminary testing on alkenes of different substitution patterns (terminal, 1,1-, tri-substituted) to see if this deuteroazidation can behave well over these representative examples. Excitingly, deuteroazidation isotopologues could be prepared using a similar catalytic system, with adjustment of iron salt and increased light intensity, allowing product formation in moderate yields with good deutero-incorporation rate (80–89% D). These initial results suggest the feasibility of this approach and offer a synthetically useful solution to deuterated alkyl azides.

### Scope of regioselective chloro- and bromo-azidation of alkenes

Encouraged by the broad scope of our photocatalytic hydroazidation, we sought to investigate if this iron-azide LMCT process can cooperate with other reaction schemes, such as using electrophilic halogen sources to permit XAT and achieve photocatalytic regioselective haloazidation, which would provide an efficient route to diverse alkyl azides bearing halide reaction handles. In our hypothesis, an ideal electrophilic halogen source could not only deliver halogen functionality efficiently and regioselectively via homolytic XAT, but also serve as single electron oxidant that could reoxidize lower valent iron to its higher valent photoactive state for the next LMCT catalysis cycle. Thus, this tandem LMCT/XAT could provide a competitive strategy of redox-neutral, regioselective haloazidation. Notably, while pioneering studies by Finn[24], Kappe[25], Feng[26], Burns[27], and coworkers have provided intriguing approaches to this transformation, the reliance on stoichiometric exogenous oxidant or molecular halogen often results in low regioselectivity control due to unselective ring opening of cationic halonium intermediates. Further, the requirement of directing group or specific substrate class for efficient reaction have rendered these protocols less general towards the development of unified strategies that accommodate broad scope of alkenes.

Tackling these challenges, we were delighted to find electrophilic N-chloro- and N-bromo-succinimides (NCS, NBS) can be efficiently leveraged in photocatalytic, regioselective chloro- and bromoazidation (Fig. 3). A broad range of unactivated alkenes (53–70; 78–90) and activated alkenes (71–74; 91–94) were converted to the corresponding haloazidated products in moderate to good yields. Reminiscent to our hydroazidation scope, simple aliphatic alkenes (53, 78) and protecting groups including benzoate (54, 55, 79), tosylate (80), and 2,2,2-trichloroethoxycarbonyl (Troc) (82) afforded corresponding chloroazidation and bromoazidation products efficiently. Unactivated alkenes containing heterocycles that are commonly used in structure-activity relationship studies were also compatible with our chloroazidation and bromoazidation protocols. Sensitive functionalities including sulfonamides (56) and benzamides (57) with acidic N–H protons, along with N-methylated sulfonamides (81) and strained rings (61, 62, 87), underwent smooth chloro- and bromoazidation without compromising structural integrity and labile functionalities that are prone to nucleophilic substitution (63, 86) and oxidation or reduction (64–67, 85) were individually examined to explore the generality of our protocols, furnishing difunctionalized products in moderate to good yields. The encouraging compatibility with these sensitive and redox-labile functionalities further showcases the synthetic advantage of our system over previous protocols that were dependent on strong exogenous oxidants and molecular halogen, where the decomposition of these functionalities would be expected. Substitution patterns were also investigated, demonstrating tolerance of 1,1-disubstituted alkenes (68, 69, 88, 89) and sterically hindered tri-substituted alkenes (70, 90). Importantly, these difunctionalizations indicate generality to activated alkenes including styrene-type (71, 91) and alkenes adjacent to heteroatom functionality (72–74; 92–94), with these entries all delivering the chloro- and bromo- azidation products smoothly. Similar to the hydroazidation protocol, these haloazidation conditions can also be employed on olefins containing API fragments (75, 76, 95), agrochemical moieties (96), and a natural product (77), which allows the simultaneous incorporation of two (pseudo)halides reaction handles into bioactive compounds for chemical biology studies. The versatility and broad compatibility of these protocols towards various functional groups and low-toxicity of the earth-abundant iron catalyst used in these haloazidation systems further demonstrate cooperative LMCT/XAT as a powerful approach to regioselective heterodifunctionalization of olefins.

### Scalability, late-stage application, and mechanistic studies

Bolstered by the extensive scope of our hydroazidation and haloazidation protocol, we further sought to demonstrate the scalability and late-stage application of the alkyl azide products. First, large scale hydroazidation was performed, giving corresponding product in good yield. We then sought to test if our protocol could provide entry to a simple late-stage transformation. Following facile chloroazidation, we performed a simple work-up (no isolation required) to remove the iron salt and directly employed the crude chloroazidation product for subsequent Huisgen cyclization, affording synthetically useful yield of chloro-substituted triazole (Fig. 4A). Next, we sought to gain preliminary insight into the mechanism of this reaction. First, including 1 equivalent of 2,2,6,6-tetramethyl-1-piperidinyloxy (TEMPO), a common radical scavenger, in our optimized conditions for alkene hydroazidation resulted in complete suppression of reactivity and almost full recovery of alkene, indicating a radical process is likely (Fig. 4B). With this result in hand, we applied radical clock substrates in both hydroazidation and chloroazidation to further provide support for a radical pathway. As expected, 5-exo-trig ring-closing products are obtained in both entries, which suggests the subsequent HAT or XAT step is slower than the rate constate of $2 \times 10^5$ (approx. for 5-exo-trig) and further supports the system should proceed via a radical mechanism (Fig. 4C). To provide more kinetic details of this cooperative system, we then interrogated the kinetic isotope effect of our hydroazidation[69], where a moderate primary KIE value of 2.0 was indicated, suggesting the HAT could have rate determining character in this cooperative system (Fig. 4D) as observed in alkene hydrofluoroalkylation[63]. To investigate whether the ligand could affect the spectroscopic character of the key photoactive species, we first performed UV–Vis studies of the combination of iron salts and ligand. Unligated iron-azide shows minimal absorption under 400 nm and almost no absorption above 400 nm, consistent with the low reactivity observed using iron catalyst alone. By contrast, the combination of Fe(III) and terpyridine ligand offers a bathochromic shift, which correlates with our observation that ligated iron demonstrates higher reactivity when using wavelengths above 400 nm. Time-resolved experiments demonstrate gradual attenuation of the charge transfer peak of iron(III)-azide, followed by emergence of absorption at 550 nm, which we postulate to correspond to the characteristic absorption of the bis(terpyridine) iron(II) complex (for more details, see supporting information). Based on these results and pioneering work from Zuo and coworkers[64], we reason that ligand enhances the reactivity by improving the bathochromic absorption overall, facilitating a more productive light-induced homolysis process.

In summary, we have demonstrated a general photocatalytic protocol for anti-Markovnikov hydroazidation of alkenes enabled by

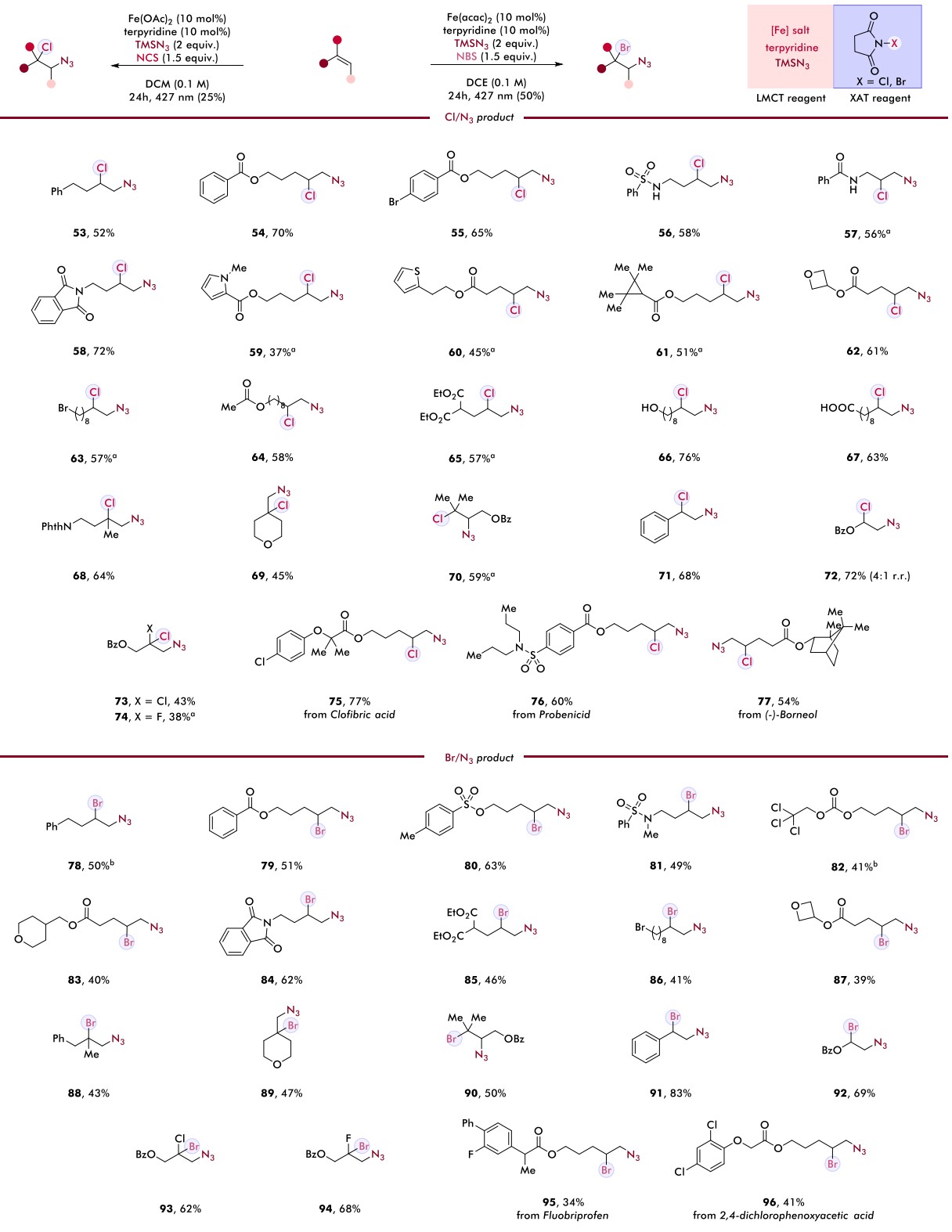

**Fig. 3 | Scope of haloazidation of alkenes.** Reaction conditions of chloroazidation of alkene (0.1 mmol, 1.0 equiv.), TMSN$_3$ (2.0 equiv.), Fe(OAc)$_2$ (10 mol%), terpyridine (10 mol%), NCS (1.5 equiv.) and DCM (0.1 M), 24 h, RT, 427 nm Kessil LED (25%). Reaction conditions of bromoazidation of alkene (0.1 mmol, 1.0 equiv.), TMSN$_3$ (2.0 equiv.), Fe(acac)$_2$ (10 mol%), terpyridine (10 mol%), NBS (1.5 equiv.) and DCE (0.1 M), 24 h, RT, 427 nm Kessil LED (50%).[a] 48 h. [b] Determined by NMR using CH$_2$Br$_2$ as internal standard.

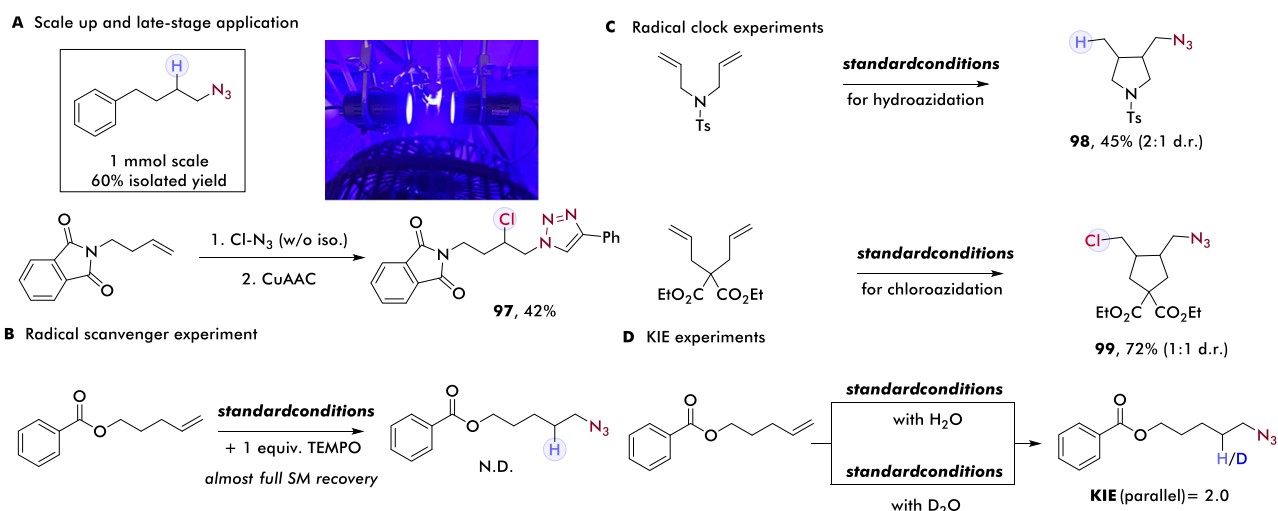

**Fig. 4 | Late-stage application and mechanistic studies.** **A** Late-stage application of chloroazidation product. **B** Radical scavenger experiment. **C** Radical clock experiments. **D** KIE experiments. CuAAc copper-catalyzed azide-alkyne Huisgen cyclization.

cooperative LMCT and HAT. Broad scope, mild visible light conditions, and late-stage modification of pharmaceuticals/natural products derivatives were illustrated, addressing substrate limitations in previous anti-Markovnikov hydroazidation methods via radical chain pathways or stoichiometric metal-photochemical pathways. Critical to this transformation is coordination of earth-abundant iron with tridentate ligand, facilitating azidyl radical generation under visible light irradiation, and compatibility with organic thiol cocatalyst in efficient HAT to access diverse alkyl azides and some representative deuterated analogues in high anti-Markovnikov regioselectivity. With judicious choice of halogenating reagents, we were able to develop a complementary tandem LMCT/XAT reaction manifold to access redox-neutral, regioselective haloazidation of a broad range of olefins, addressing previous challenges of regioselectivity control in the absence of directing groups and specific alkene classes. We expect these cooperative strategies can serve as a powerful tool in developing redox-neutral hydro- and difunctionalization of feedstock chemicals and future studies based on this dual catalytic manifold are ongoing in our laboratory.

## Method

### General procedure of photocatalytic hydro-, deuteroazidation of alkenes

Fe salt (10 mol%, 0.1 equiv.) and terpyridine (10 mol%, 0.1 equiv.) were added in an oven-dried 8-mL test vial containing a Teflon®-coated magnetic stir bar. The vial was evacuated and backfilled with $N_2$ (repeated for 4 times), followed by addition of alkenes (0.1 mmol, 1.0 equiv.), $TMSN_3$ (0.40 mmol, 4.0 equiv.), 4-F-thiolphenol (10 mol%, 0.1 equiv.) in $HCCl_3/H_2O$ (19:1, 0.1 M in regard to alkenes) via syringe under $N_2$. For deuteroazidation of alkenes, $HCCl_3/D_2O$ is used (19:1, 0.1 M in regard to alkenes). The reaction mixture was placed under 427 nm Kessil® light after sealing the punctured holes of the vial cap with vacuum grease and electric tape/parafilm for better air-tight protection and allowed to react at room temperature for 36–96 h. Following this, the reaction mixture was filtered through a pad of celite, which was subsequently rinsed with DCM. The filtrate was concentrated, and the residue was then purified by flash column chromatography or preparatory thin-layer chromatography to give the corresponding hydroazidated products. The details of hydro- and deuteroazidation of alkenes (the types and equivalents of iron salt, light intensity, and time in each protocol) are demonstrated in the Supplemental Information 1.3.

### General procedure of photocatalytic haloazidation of alkenes

Fe salt (10 mol%, 0.1 equiv.) and terpyridine (10 mol%, 0.1 equiv.) were added in an oven-dried 8-mL test vial containing a Teflon®-coated magnetic stir bar. The vial was evacuated and backfilled with $N_2$ (repeated for 4 times), followed by addition of alkenes (0.1 mmol, 1.0 equiv.), $TMSN_3$ (0.20 mmol, 2.0 equiv.), NCS or NBS (0.15 mmol, 1.5 equiv.) in DCM (for chloroazidation) or DCE (for bromoazidation) (0.1 M in regard to alkenes) via syringe under $N_2$. The reaction mixture was placed under 427 nm Kessil® light after sealing the punctured holes of the vial cap with vacuum grease and electric tape/parafilm for better air-tight protection and allowed to react at room temperature for 24 h. Following this, the reaction mixture was filtered through a pad of celite, which was subsequently rinsed with DCM. The filtrate was concentrated, and the residue was then purified by flash column chromatography or preparatory thin-layer chromatography to give the corresponding haloazidated products. The details of haloazidation of alkenes (the types and equivalents of iron salt, light intensity, and time in each protocol) are demonstrated in the Supplemental Information 1.3.

## Data availability

The authors declare that all the data supporting the findings of this research are available within the article and its supplementary information. Data supporting the findings of this manuscript are also available from the corresponding author upon request.

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

## Acknowledgements

We acknowledge financial support from NIH (R35GM142738), the Welch Foundation (C-2085), RCSA (CS-CSA-2023-007), and ACS-PRF (62397-DNI1). Dr. Yohannes H. Rezenom (TAMU/LBMS), Dr. Ian M Riddington (UT Austin Mass Spectrometry Facility), and Dr. Christopher L. Pennington (Rice University Mass Spectrometry Facility) are acknowledged for assistance with mass spectrometry analysis.

## Author contributions

K.-J.B. and J.G.W. designed the project. K.-J.B., S.Y., Y.C., Q.L., X.-W.C., D.N.Jr, and S.-C.K. performed the experiments. K.-J.B and J.G.W. wrote the manuscript. J.G.W. directed the project. K.-J.B., S.Y., Y.C., Q.L., X.-W.C., D.N.Jr, S.-C.K., A.A.M., and J.G.W. interpreted the results in the manuscript.

## Competing interests

The authors declare no competing interests.
