## [Transparent Peer Review file · Nature Communications]

Photocatalytic Anti-Markovnikov Hydro- and Haloazidation of Alkenes

Corresponding Author: Professor Julian West

Version 0:

Reviewer comments:

Reviewer #1

(Remarks to the Author)

The authors demonstrated 3 photocatalytic iron catalyzed systems for anti-Markovnikov hydroazidation, chloroazidation and bromoazidation of alkenes using TMSN₃ as azido source. The key steps they proposed are light promoted LMCT of Fe(III)-N₃ to form azidyl radical, addition of azidyl radical to olefin, H/Cl/Br atom transfer and oxidation of Fe(II) to Fe(III) by thiyl radical or NXS based oxidants. These reactions display excellent regioselectivity (Anti-Markovnikov). The mild conditions overcome some limitations of previous anti-Markovnikov hydroazidation methods via radical mechanism. In most cases, the yields are moderate. This work is an important progress of in the field of LMCT chemistry of iron developed recently, especially, the identification of the terpyridine ligand in this chemistry. Over all I would like to recommend the publication of this work in this journal after addressing the following issues:

- 1, since the use of terpyridine is pivotal, the identification of the terpyridine ligated Fe(III)N₃ and studying its photochemical properties should be done. (recently, Zuo group reported such complexes (jacs asap), and should be cited.
- 2, For the haloazidation, the control experiments should be carried out and discussed. I wonder if the combination of TMSN₃/NXS or TMSN₃/NXS/Fe(III) (cat) might promote the same conversions, but the regioselectivity might be an issue.
- 3, The use of Fe(OAc)₂ for chloroazidation and Fe(asac)₂ for bromoazidation is quite interesting, any comments?
- 4, according the mechanism, alkyne is possible candidate. Have the authors ever tried this? It would be great to obtain the vinyl azide. Conjugated dienes are also recommended to be included in the substrate.
- 5, Scale-up experiments are recommended, for example gram-scale.
- 6, some errors: in Figure 2, for most case, isolated yields are reported? 'HN₃ product' should be 'H/N₃ product', note d is for compound 31, not for 45, right? The arrangement of 40-41 need improvement for reader-friendly, numbers and structures separate too far away. The remained double bond in structure 31 is not an activated double bond, in contrast it is a deactivated double bond in the context of electrophilic azidyl radical. So the related comment on the selectivity should be adjusted to avoid misleading. Figure 3, top right, the information in the red square is not correct, I think. Its footnote repeats chloroazidation.

There are several typos in the text. Some errors in the supplementary information need to be corrected, such as Line spacing, subscript and solvent expression in the initial testing form. There are not page numbers.

- 7, Some HNMR spectra need explanation for clarity, such as the third NMR spectrum of compound 9, to show the regioselectivity?

Reviewer #2

(Remarks to the Author)

In this paper, West and coworkers developed an efficient photocatalytic strategy for hydro- and haloazidation of alkenes by integrating LMCT/HAT or LMCT/XAT. The redox-neutral process exhibited mild conditions and high anti-Markovnikov regioselectivity. A nice substrate scope for activated/unactivated alkenes is presented. The corresponding products are shown clearly and supported by comprehensive characterization data. This methodology provides a powerful tool for olefin hydro- and di-functionalization. The comparison with existing methods and the demonstration of the method's broad substrate scope and functional group tolerance highlight its potential impact on related fields. However, more evidence should be provided to support some of the claims in the mechanism study part. Therefore, I think this paper can meet the high standard of Nature Communications after addressing these issues:

1. The mentioned red shift in UV-Vis spectra of the iron complex with or without the ligand is not presented in the mechanism

study part, even the note of Figure 4 contains the corresponding part (Figure 4E/F). The UV-Vis experiments are important evidence for the effects of the ligand. Please add them.

2. Step A in Figure 1D seems to be a multi-step process. Investigation of CV of thiophenol, iron w/ ligand, and iron w/o ligand may give more insight into the SET process.

3. The author claims that the ligand can accelerate the LMCT process. Thus some kinetic experiments with a simplified probe reaction should be performed to compare the homolysis rate w/ or w/o the ligand.

Minor points:

1. As regards entry 8 of Table 1, the reaction time is 36 h as well. So what is the meaning of the sentence "extended reaction time..." in the text? Please verify it.

2. On Page 9, Line 12, the sentence "Lastly, control experiments..." seems confusing. Please check.

3. On page 12, line 10, the number (8) should be attributed to "6" according to Figure 2. Similarly on page 14, line 6, the number (37) should be attributed to "41". Please recheck the whole part.

4. The annotation at the bottom of Figure 2 is reduplicative with the annotation in the note of Figure 2. A similar issue also exists in Figure 3.

5. An unreasonable discrepancy between the calculated and actual value appears for the HRMS data of compound 5. Please verify it.

6. Some prepositions, prefixes, and capital letters of chemical names should be in italics, please check and revise them.

7. Some other examples of photoinduced LMCT with transition metals should be cited (e.g. Science 2018, 361, 668; Green Chem., 2021, 23, 6984; Nat. Commun., 2024, 15, 8619).

Reviewer #3

(Remarks to the Author)

The authors reported a Photocatalytic Anti-Markovnikov Hydro- and Haloazidation of Alkenes. Although Anti-Markovnikov hydroazidation of alkenes has been investigated by various research groups using different reagent systems and transition metal catalysts, the reaction still faces significant challenges, such as substrate limitations. The reviewer find some novelty in the given strategy, thus, I suppose that this manuscript could become suitable for publication in Nature Communications after revision addressing several concerns which are listed below:

1. The scope presented in Figure 2 contains redundant examples of structurally related alkenes. I suggest adding some other results at least for tetrasubstituted alkenes, conjugated dienes, N-protected indoles. Unsuccessful results should also be mentioned.

2. The SI should indicate some physical properties of the products, at least the melting point and color.

3. Some compound's spectra show obvious impurities and require further purification. If purification is not possible, at least indicate the composition of impurities.

Version 1:

Reviewer comments:

Reviewer #1

(Remarks to the Author)

The authors have addressed my questions with high quality, I recommend to published as it.

Reviewer #2

(Remarks to the Author)

The comments raised by this reviewer have been well addressed by the authors. The manuscript can be accepted in its current form.

Reviewer #3

(Remarks to the Author)

The authors reported a Photocatalytic Anti-Markovnikov Hydro- and Haloazidation of Alkenes. West and coworkers have provided a positive response to the questions I raised and have also corrected some minor issues in the SI section. Therefore, I think this paper can meet the high standard of Nature Communications.

Reviewer #1 (Remarks to the Author):

The authors demonstrated 3 photocatalytic iron catalyzed systems for anti-Markovnikov hydroazidation, chloroazidation and bromoazidation of alkenes using TMSN₃ as azido source. The key steps they proposed are light promoted LMCT of Fe(III)-N₃ to form azidyl radical, addition of azidyl radical to olefin, H/Cl/Br atom transfer and oxidation of Fe(II) to Fe(III) by thiyl radical or NXS based oxidants. These reactions display excellent regioselectivity (Anti-Markovniko). The mild conditions overcome some limitations of previous anti-Markovnikov hydroazidation methods via radical mechanism. In most cases, the yields are moderate. This work is an important progress of in the field of LMCT chemistry of iron developed recently, especially, the identification of the terpyridine ligand in this chemistry. Over all I would like to recommend the publication of this work in this journal after addressing the following issues:

We are grateful for the reviewer's enthusiasm for our work and excited to receive this recognition. We appreciate the detailed suggestions given by the reviewer!

1) since the use of terpyridine is pivotal, the identification of the terpyridine ligated Fe(III)N₃ and studying its photochemical properties should be done. (recently, Zuo group reported such complexes (jacs asap), and should be cited.

We thank the reviewer for pointing out this important reference! The pioneering work from Zuo and coworkers on unique reactivity of iron/terpyridine complex for LMCT process is truly inspirational and we have added this paper as Ref. 64 in the revised manuscript. Additionally, we have added the following note in the manuscript (highlighted) to showcase the importance of this work, which is also shown as follows,

Zuo and coworkers reported a pioneering study of iron terpyridine catalysis utilizing LMCT to achieve highly efficient aerobic carbonylation of methane, showcasing the unique photochemical properties of ligated iron complexes in LMCT processes.

2. For the haloazidation, the control experiments should be carried out and discussed. I wonder if the combination of TMSN₃/NXS or TMSN₃/NXS/Fe(III) (cat) might promote the same conversions, but the regioselectivity might be an issue.

We thank the referee for this important question! As suggested, we have performed control experiments to see if the combination of TMSN₃/NXS or the conditions without light could promote the conversions. The results are shown below and we have also added these screenings to the optimization table in the section '2.2 Optimization of chloroazidation' and '2.3 Optimization of bromoazidation' in the revised supporting information,

For chloroazidation, we have performed more control experiments to demonstrate the indispensability of iron, ligand and light. When using Fe(NO₃)₃·9H₂O as the catalyst, the reaction showed moderate reactivity without light, while this ground state reactivity was less efficient when using Fe(OAc)₂ as catalyst. Notably, the ground state reactivity for iron salts was only observed when using iron with ligand, indicating the unique characteristics of terpyridine-ligated Fe(III) azide species.

entry	iron salt	ligand	light	yield
1	Fe(NO ₃) ₃ ·9H ₂ O	terpyridine	427% (25%)	80
2	Fe(NO ₃) ₃ ·9H ₂ O	terpyridine	dark	48
3	Fe(NO ₃) ₃ ·9H ₂ O	no ligand	dark	ND
4	Fe(OAc) ₂	terpyridine	427% (25%)	72
5	Fe(OAc) ₂	terpyridine	dark	12

Similarly, we have also performed series of control experiments for bromoazidation. As the reviewer suggested, we observed significant amount of the regioisomeric bromoazide when using [Fe]/L in the absence of blue light. No product was obtained in the absence of iron and ligand.

Indeed, this low regioselectivity under ground state conditions was observed when using other iron salts and substrates, suggesting the importance of photochemical pathway for achieving high regioselectivity. We have also provided crude NMR and isolated product for entry 2 to showcase this result.

entry	iron salt	ligand	light	yield
1	Fe(NO ₃) ₃ ·9H ₂ O	terpyridine	427% (25%)	48
2	Fe(NO ₃) ₃ ·9H ₂ O	terpyridine	dark	68 (4:1 r.r.)
3	no iron	no ligand	427% (25%)	ND
4 ^a	Fe(OAc) ₂	terpyridine	427% (25%)	52
5 ^a	Fe(OAc) ₂	terpyridine	dark	36 (1.3:1 r.r.)

^a 10 mol% of Fe(OAc)₂, ligand and 2.0 equiv. of TMSN₃.

it) Kangjie 17

We really appreciate the reviewer for this suggestion and these findings also indicate pivotal importance of Fe-terpy combination to achieve high reactivity and light activation to ensure high regioselectivity.

3. The use of Fe(OAc)₂ for chloroazidation and Fe(acac)₂ for bromoazidation is quite interesting, any comments?

We thank the referee for noting this interesting point! We have consistently observed intriguing counterion effects in Fe LMCT photocatalysis, with different reactions proceeding most effectively with different salts. These effects are challenging to deconvolute completely, though we propose that key properties include lability of the counterion as a ligand helping to control the coordination sphere, basicity of the counterion modulating reaction pH, redox activity of the counterion modulating Fe oxidation state, and solubility of the salt in the reaction medium. Additionally, the starting oxidation state of Fe(II or III) could impact the concentration of photoactive Fe(III) species over the course of the reaction and relative abundance of counterions.

Basicity, ligand lability, and solubility seem to be the most likely differences between Fe(OAc)₂ and Fe(acac)₂ in the context of Haloazidation, though the impact of these differences appears to be small. Indeed, we noted that both Fe(OAc)₂ and Fe(acac)₂ formed bromoazidation products in similar yields during our initial optimization (SI Section 2.3 – Iron Screening table); however, Fe(acac)₂ exhibited slightly improved mass balance, leading us to proceed with this salt.

We saw a much larger difference between Fe(OAc)₂ and Fe(NO₃)₃·9H₂O in both haloazidation reactions, with significant diazidation product being observed when Fe(NO₃)₃·9H₂O is used. This observation is consistent with our previous work on photocatalytic diazidation (*Chem. Sci.*, **2024**, *15*, 124-133), where we found Fe(NO₃)₃·9H₂O to be especially effective for alkene diazidation. We hypothesize that the main factors for this heightened diazidation activity are the high lability of NO₃⁻, allowing for Fe(III) polyazides proposed to enable diazidation (e.g. *ACS Catal.* **2018**, *8*, 4473–4482) to be more easily formed, and the higher starting concentration of Fe(III) due to the salt oxidation state, similarly increasing polyazide concentration.

4. according the mechanism, alkyne is possible candidate. Have the authors ever tried this? It would be great to obtain the vinyl azide. Conjugated dienes are also recommended to be included in the substrate.

We thank the reviewer for this great suggestion! Attempting the reactions with alkynes led to no desired product formation and only substrate was recovered. We hypothesize that further screenings of different reaction conditions including different thiol sources may offer a solution, though this has thus far been unsuccessful.

Excitingly, reaction of a conjugated diene affords the corresponding product. The result is provided here for reviewers' convenience and also added in the section '2.4 Additional substrates for photocatalytic hydroazidation' in the revised supporting information.

¹H NMR (600 MHz, CDCl₃) δ 8.06 – 8.01 (m, 2H), 7.59 – 7.54 (m, 1H), 7.48 – 7.40 (m, 2H), 5.84 – 5.72 (m, 0.37H), 5.67 – 5.53 (m, 1.45H), 5.34 – 5.27 (m, 0.18H), 4.40 – 4.30 (m, 2H), 4.02 – 3.67 (m, 0.60H), 3.33 – 3.18 (m, 1.26H), 2.58 – 2.25 (m, 3H), 1.93 – 1.68 (m, 1H).

¹³C NMR (151 MHz, CDCl₃) δ 166.55, 135.34, 132.92, 130.35, 129.56, 128.83, 128.79, 128.38, 128.36, 127.90, 123.95, 64.13, 52.70, 50.94, 32.23, 32.12, 28.76, 28.17.

HRMS ESI: [M+Na]⁺ calcd. for C₁₃H₁₅N₃O₂Na: 268.1056; Found 268.1058

Besides conjugated alkenes, we have also tested our protocol on tetrasubstituted alkenes suggested by the reviewers and the encouraging results are shown below. We really appreciated the suggestions from the reviewers to help us to further test the tolerance of our protocol and we have added this example to the revised Figure 2 as compound 31 in the manuscript.

We have also added the following note to suggest the compatibility with sterically hindered alkenes,

More sterically hindered tetrasubstituted alkene (31) also gave good yield of corresponding product in a mixture of both regioisomers.

5. Scale-up experiments are recommended, for example gram-scale.

We appreciate the referee for raising this important point! Considering potential explosive hazard of large-scale reaction, we used 1 mmol scale to showcase the relative scalability, and the result has been added to the revised Figure 4 in the manuscript and shown below for the convenience of referee,

We have also added the following note to the revised manuscript (highlighted),

we further sought to demonstrate the scalability and late-stage application of the alkyl azide products

...

First, large scale hydroazidation was performed, giving corresponding product in good yield.

6. some errors: in Figure 2, for most case, isolated yields are reported? 'HN3 product' should be 'H/N3 product', note d is for compound 31, not for 45, right? The arrangement of 40-41 need improvement for reader-friendly, numbers and structures separate too far away. The remained double bond in structure 31 is not an activated double bond, in contrast it is a deactivated double bond in the context of electrophilic azidyl radical. So the related comment on the selectivity should be adjusted to avoid misleading. Figure 3, top right, the information in the red square is not correct, I think. Its footnote repeats chloroazidation. There are several typos in the text. Some errors in the supplementary information need to be corrected, such as Line spacing, subscript and solvent expression in the initial testing form. There are not page numners.

We thank the referee for spotting these errors! For all products in Figure 2, the isolated yields are given unless noted. We have also comprehensively revised Figures 2 and 3 (revising the superscript, reorganization for Figure 2 and correcting the reaction conditions information for Figure 3) to improve the clarity and adjust the related comments as the referee suggested!

For structure 31, we have adjusted the description (highlighted) according to reviewer's suggestion,

...provided reaction at the non-conjugated site in exclusive anti-Markovnikov regioselectivity.

We have also corrected the typos including line spacing, subscript, solvent information in the optimization table and added page numbers to the SI. We thank the referee for pointing out this important information and helping us improve the quality of this work!

7. Some HNMR spectra need explanation for clarity, such as the third NMR spectrum of compound 9, to show the regioselectivity?

We thank the reviewer for raising this point! The third spectrum indicates the regioselectivity of all the hydroazidation products where we have also added the regioselectivity information in the individual characterization of hydroazidation products to the section '2.3 Characterization of Corresponding Products'. To provide more clarity, we have added a note in section 'III. Supplemental Figures',

The NMR spectra consisted of ^1H NMR, ^{13}C NMR and regioselectivities of isolated products.

We hope this response can address the reviewer's comments. We thank the reviewer for their dedication to this revision and for helping us improve the quality of this work!

Reviewer #2 (Remarks to the Author):

In this paper, West and coworkers developed an efficient photocatalytic strategy for hydro- and haloazidation of alkenes by integrating LMCT/HAT or LMCT/XAT. The redox-neutral process exhibited mild conditions and high anti-Markovnikov regioselectivity. A nice substrate scope for activated/unactivated alkenes is presented. The corresponding products are shown clearly and supported by comprehensive characterization data. This methodology provides a powerful tool for olefin hydro- and di-functionalization. The comparison with existing methods and the demonstration of the method's broad substrate scope and functional group tolerance highlight its potential impact on related fields. However, more evidence should be provided to support some of the claims in the mechanism study part. Therefore, I think this paper can meet the high standard of Nature Communications after addressing these issues:

We are grateful for the recognition and the support of this referee for our work!

1. The mentioned red shift in UV-Vis spectra of the iron complex with or without the ligand is not presented in the mechanism study part, even the note of Figure 4 contains the corresponding part (Figure 4E/F). The UV-Vis experiments are important evidence for the effects of the ligand. Please add them.

We thank the reviewer for the suggestion! In the revision, we have performed the UV-Vis studies and time-resolved studies to showcase the spectroscopic details of ligated iron complex. We have also added this part of the study in the section '2.7 Spectroscopic studies of different iron species and time-resolved UV-visible studies' in the revised supporting information. The details are shown below for the convenience of referee,

Left:

(a) Black line: $\text{Fe}(\text{NO}_3)_3 \cdot 9\text{H}_2\text{O}$ (0.01 mmol) and TMSN_3 (0.40 mmol) were added to HCCl_3 (0.1 M, 1 ml), offering a 10 mM solution, followed by dilution to 1 mM in HCCl_3 and transferred to UV-visible cell for measurement.

(b) Blue line: FeCl_2 (0.01 mmol), terpyridine (0.01 mmol) and TMSN_3 (0.40 mmol) were added to HCCl_3 (0.1 M, 1 ml), offering a 10 mM solution, followed by dilution to 1 mM in HCCl_3 and transferred to UV-visible cell for measurement.

(c) Red line: $\text{Fe}(\text{NO}_3)_3 \cdot 9\text{H}_2\text{O}$ (0.01 mmol), terpyridine (0.01 mmol) and TMSN_3 (0.40 mmol) were added to HCCl_3 (0.1 M, 1 ml), offering a 10 mM solution, followed by dilution to 1 mM in HCCl_3 and transferred to UV-visible cell for measurement.

Right:

$\text{Fe}(\text{NO}_3)_3 \cdot 9\text{H}_2\text{O}$ (0.01 mmol) was added in an oven-dried 8-mL test vial containing a Teflon®-coated magnetic stir bar. The vial was evacuated and backfilled with N_2 (repeated for 4 times), followed by addition of cyclohexane (0.1 mmol) and TMSN_3 (0.40 mmol) in HCCl_3 (1 ml) via syringe under N_2 . The 20 μl of solution mixture was transferred to a capped UV-visible cell (containing 2 ml HCCl_3) via syringe and degassed the solution for 2 minutes. The UV-visible cell was irradiated by 390nm Kessil® light using irradiation time intervals of 3, 8, 18, 80, 150, 300, 400 s. After each irradiation, the cell was vigorously shaken for 15 sec, ensuring good mixing of the solution, followed by performing UV-visible characterization at respective intervals.

In the spectroscopic studies of different iron species, low absorption is observed below 400 nm and almost no absorption was observed above 400 nm, correlating with the control experiments that the reaction exhibited low efficiency when using 390 nm and almost no reactivity when using 427 nm without ligand. When using ligated iron, a significant bathochromic shift is indicated, providing support that the ligated system gave higher efficiency when using 427 nm light.

Additionally, we have also performed time-resolved UV-Vis studies to further interrogate the photoactive species in the light-induced homolysis process. Time-resolved experiments demonstrated gradual attenuation of charge transfer peak, and absorption at ~ 550 nm indicates characteristic of the bis(terpyridine) iron(II) complex which we also used Fe(II) salt under analogous conditions (A, green line) and correlates with Zuo and coworkers' findings (Ref. 64), suggesting the generation of ligated Fe(II).

We have also added the following notes to 'Scalability, late-stage application and mechanistic studies' in the revised manuscript (highlighted) to demonstrate these spectroscopic details of our system,

To investigate whether the ligand could affect the spectroscopic character of the key photoactive species, we first performed UV-Vis studies of the combination of iron salts and ligand. Unligated iron-azide shows minimal absorption under 400 nm and almost no absorption above 400 nm, consistent with the low reactivity observed using iron catalyst alone. By contrast, the combination of Fe(III) and terpyridine ligand offers a bathochromic shift, which correlates with our observation that ligated iron demonstrates higher reactivity when using wavelengths above 400 nm. Time-resolved experiments demonstrate gradual attenuation of the charge transfer peak of iron(III)-azide, followed by emergence of absorption at 550nm, which we postulate to correspond to the characteristic absorption of the bis(terpyridine) iron(II) complex (for more details, see supporting information). Based on these results and pioneering work from Zuo and coworkers,⁶⁴ we reason that ligand enhances the reactivity by improving the bathochromic absorption overall, facilitating a more productive light-induced homolysis process.

We really appreciate the reviewer's comments prompting us to perform these studies and helping us to improve the quality of our work.

2. Step A in Figure 1D seems to be a multi-step process. Investigation of CV of thiophenol, iron w/ ligand, and iron w/o ligand may give more insight into the SET process.

We thank the reviewer for this suggestion! We have carried out Cyclic Voltammetry studies of thiol and iron both with and without ligand to share more insights into the SET process. We show these data below and have also added this part of studies as the section '2.8 Cyclic Voltammetry studies of photocatalytic hydroazidation' in the revised supporting information.

(a) CV Analysis of ferrocene (3 mM in MeCN or CHCl_3).

(b) CV Analysis of $\text{Fe}(\text{OAc})_2$, $\text{Fe}(\text{OAc})_2$ with terpy ligand, 4-fluorothiophenol, Sodium 4-fluorothiophenolate (3 mM in MeCN).

(c) CV Analysis of $\text{Fe}(\text{OAc})_2$, $\text{Fe}(\text{OAc})_2$ with terpy ligand (3 mM in EtOH).

Cyclic voltammetry (CV) was performed in a three-electrode system using a CHI 680D electrochemical workstation (CHI Instruments, USA). The working electrode was a glassy carbon disk electrode (diameter: 5.0 mm, PTFE shroud). A platinum wire served as the auxiliary electrode, and an Ag/AgCl electrode (saturated KCl solution) was used as the reference electrode. Electrolyte solutions consisted of either 0.1 M tetrabutylammonium tetrafluoroborate (NBu_4BF_4) in organic solvents (acetonitrile, ethanol, and chloroform) or 0.1 M KCl in water. The CV measurements were conducted at a scan rate of 50 mV s^{-1} . Prior to sample analysis, the solvent window was scanned to confirm the absence of electroactive impurities. Afterward, ferrocene (Fc, 3 mM) was introduced as an internal standard to calibrate the redox potential. Before measurements, the glassy carbon electrode was polished with $0.05 \mu\text{m}$ alumina slurry and thoroughly rinsed with deionized water and acetone.

The difficulty in capturing redox behavior in biphasic systems led us to explore the CV in monophasic solutions. Limited solubility of different iron salts and terpyridine ligand in chloroform led to less obvious redox peaks when carrying out our initial tests, leading us to use acetonitrile and ethanol to compare well-defined redox profiles of thiol and iron with and without ligand (Figure A-C).

In Figure B, iron w/ ligand exhibits a redox potential at -0.65 V vs Fc^+/Fc in MeCN and the $\text{PhS}^\bullet/\text{PhS}^-$ -couple gives -0.45 V vs Fc^+/Fc in MeCN, suggesting reduction of thiyl radical to thiolate by $\text{Fe}(\text{II})$ is possible, even allowing for the variation in potentials across solvents (*Inorg. Chim. Acta* **2000**, 298 (1), 97–102). Additionally, previous references studying the redox couple of iron and thiol also indicates this redox process is favorable (*J. Phys. Chem.* **1996**, 100 (23), 9892–9899), providing further support that outer-sphere transfer between the two intermediates is thermodynamically feasible.

In Figure C, the Fe(III/II) redox couple for Fe(II) without ligand has been measured to be -0.63 V vs Fc+/Fc in ethanol and -0.65 V vs Fc+/Fc for Fe(II) with ligand. These results suggest the addition of terpyridine has minimal effect of tuning the redox potential of Fe(III/II) in this case.

We appreciate this important suggestion from the reviewer which has helped us to gain more insight into our system!

3. The author claims that the ligand can accelerate the LMCT process. Thus some kinetic experiments with a simplified probe reaction should be performed to compare the homolysis rate w/ or w/o the ligand.

We thank the referee for raising this important point! While the addition of ligand dramatically improves the reactivity, we note that ‘the acceleration of LMCT process’ is not necessarily an accurate statement given our observations. As shown in the optimization tables in the manuscript and supporting information, reaction efficiency is improved when using the ligand whereas trace amount of product is observed when carrying the reaction in the absence of ligand.

Since low reactivity is observed (only trace product after 36h) when the reaction is performed in the absence of ligand compared with high reactivity of standard conditions (with ligand), we reasoned the ligand is pivotal in enabling the efficient hydroazidation, though this effect is not necessarily due to acceleration of the LMCT step.

entry	deviation from standard conditions	yield
1	none	76
2	no ligand	trace

Therefore, following the reviewer’s suggestion, we have modified this description in the revised manuscript to more precisely describe this reactivity effect (highlighted) and also summarized these modifications and data here for reviewer’s convenience as follows,

Abstract

Critical to this protocol is the use of ligand to achieve efficient visible-light induced homolysis...

Introduction

...this reaction is presence of a supporting ligand for iron, where the efficiency is improved...

Minor points:

1. As regards entry 8 of Table 1, the reaction time is 36 h as well. So what is the meaning of the sentence “extended reaction time...” in the text? Please verify it.

We thank the reviewer for pointing out this typo! The reaction in entry 8 was carried out with 427 nm (50%) blue LED for 48h, offering an improved yield of 82%. We have corrected the entry as helpfully suggested.

2. On Page 9, Line 12, the sentence “Lastly, control experiments...” seems confusing. Please check.

We thank the reviewer for spotting this! We have reworded the sentence for better demonstration in the revised manuscript (highlighted), and shown as below,

...control experiments indicated no conversion in the absence of iron or light...

3. On page 12, line 10, the number (8) should be attributed to “6” according to Figure 2. Similarly on page 14, line 6, the number (37) should be attributed to “41”. Please recheck the whole part.

We thank the referee for raising this point and giving the opportunity to recheck numbering for accuracy. We have revised and double checked the numbering in the revised manuscript!

4. The annotation at the bottom of Figure 2 is reduplicative with the annotation in the note of Figure 2. A similar issue also exists in Figure 3.

We thank the reviewer for this point! We have revised the Figures to remove these duplications.

5. An unreasonable discrepancy between the calculated and actual value appears for the HRMS data of compound 5. Please verify it.

We thank the reviewer for highlighting this discrepancy! We have reanalyzed compound 5 via APCI HRMS and the result has been updated in the revised supporting information and is shown here as well:

HRMS APCI: [M-N₂+H]⁺ calcd. for C₁₃H₁₅F₃NO₂: 274.1049; Found 274.1043

6. Some prepositions, prefixes, and capital letters of chemical names should be in italics, please check and revise them.

We thank the referee for raising this important aspect! We have used italics for some prepositions, prefixes and capital letters of chemical names and double-checked to make sure the format is correct.

7. Some other examples of photoinduced LMCT with transition metals should be cited (e.g. Science 2018, 361, 668; Green Chem., 2021, 23, 6984; Nat. Commun., 2024, 15, 8619).

We appreciate the reviewer suggesting these pioneering reports regarding photoinduced LMCT with transition metal and we have added these references to the revised manuscript as Ref. 56, 42 and 48.

We have also highlighted the following works to the reference (Ref. 46 & 49) to showcase their importance in LMCT chemistry.

Zhang, Q. *et al.* Iron-Catalyzed Photoredox Functionalization of Methane and Heavier Gaseous Alkanes: Scope, Kinetics, and Computational Studies. *Organic Letters* **24**, 1901-1906 (2022)

Crocker, M. S. *et al.* Transformative ligand effects in Fe-photocatalyzed Giese-type additions. *Chem Catalysis* **4**, 101131 (2024)

We appreciate the constructive suggestions raised by the reviewer and we thank the reviewer for their dedication to this revision and helping us improve the quality of this work!

Reviewer #3 (Remarks to the Author):

The authors reported a Photocatalytic Anti-Markovnikov Hydro- and Haloazidation of Alkenes. Although Anti-Markovnikov hydroazidation of alkenes has been investigated by various research groups using different reagent systems and transition metal catalysts, the reaction still faces significant challenges, such as substrate limitations. The reviewer find some novelty in the given strategy, thus, I suppose that this manuscript could become suitable for publication in Nature Communications after revision addressing several concerns which are listed below:

We are extremely honored to receive the recognition from this reviewer for our photocatalytic anti-Markovnikov hydroazidation!

1. The scope presented in Figure 2 contains redundant examples of structurally related alkenes. I suggest adding some other results at least for tetrasubstituted alkenes, conjugated dienes, N-protected indoles. Unsuccessful results should also be mentioned.

We thank the constructive suggestion from this referee! Following the suggestion from this referee, we have carried out more scope compatibility investigations.

We observed that alkynes and N-protected indoles showed no reactivity under our standard conditions, possibly due to different reactivity profile of alkynes and aromaticity of heterocycles.

Excitingly, a tetrasubstituted alkene example showed good reactivity, giving corresponding product in 58% yield. Interestingly, conjugated dienes could also afford hydroazidation products in a ratio of 1,2- : 1,4- = 2.6 : 1 in relatively lower yield. We reason that further optimization could improve products ratio and efficiency.

We have included these results in revised Figure 2 in the manuscript (tetrasubstituted) and section '2.4 Additional substrates for photocatalytic hydroazidation' in the revised supporting information,

We really appreciate the reviewer helping us further interrogate the tolerance of our cooperative catalysis and improve the quality of this work!

2. The SI should indicate some physical properties of the products, at least the melting point and color.

We really appreciate the referee for raising this important aspect. We have added the description of product properties (melting point if solid and color of product) in the revised supporting information.

Additionally, we have provided detailed procedures (reaction conditions, isolation method, eluent and polarity) for each compound. We thank the reviewer for this important suggestion!

3. Some compound's spectra show obvious impurities and require further purification. If purification is not possible, at least indicate the composition of impurities.

We thank the reviewer for raising this point! We have purified the compounds' NMR as suggested, for example, the solvent peak of **11**, the impurities peak of **29** and **40**.

Additionally, as in most of the hydroazidation scope, there are also small amounts of inseparable Markovnikov-selective regioisomers which are not fully integrated in the ^1H NMR. With that said, we have taken the opportunity to demonstrate the regioselectivity of our reaction using primary data by adding **spectrum 3** for each products' characterization where the characteristic peaks of both regioisomers are integrated to indicate regioselectivity.

We thank the reviewer for their dedication to this revision and helping us to improve the quality of our work!

Overall, we are very grateful for the insightful suggestions from all the referees and genuinely appreciate the editor's and reviewers' dedication to help us improve the quality of our work! We hope this point-to-point response can address the suggestions from the editor and the reviewers!

Reviewer #1 (Remarks to the Author):

The authors have addressed my questions with high quality, I recommed to published as it.

We are grateful for the reviewer's enthusiasm for our work and careful help to improve our study!

Reviewer #2 (Remarks to the Author):

The comments raised by this reviewer have been well addressed by the authors. The manuscript can be accepted in its current form.

We are grateful for the reviewer's enthusiasm for our work and careful help to improve our study!

Reviewer #3 (Remarks to the Author):

The authors reported a Photocatalytic Anti-Markovnikov Hydro- and Haloazidation of Alkenes. West and coworkers have provided a positive response to the questions I raised and have also corrected some minor issues in the SI section. Therefore, I think this paper can meet the high standard of Nature Communications.

We are grateful for the reviewer's enthusiasm for our work and careful help to improve our study!